# Effects of Iron, Lime, and Porous Ceramic Powder Additives on Methane Production from Brewer’s Spent Grain in the Anaerobic Digestion Process

**DOI:** 10.3390/ma16155245

**Published:** 2023-07-26

**Authors:** Ewa Syguła, Waheed A. Rasaq, Kacper Świechowski

**Affiliations:** Department of Applied Bioeconomy, Wrocław University of Environmental and Life Sciences, 37a Chełmońskiego Str., 51-630 Wrocław, Poland; ewa.sygula@upwr.edu.pl (E.S.); waheed.rasaq@upwr.edu.pl (W.A.R.)

**Keywords:** brewers’ spent grain, biomethane production kinetics, methane fermentation, biogas, anaerobic digestion, iron powder, Fe, lime, Ca(OH)_2_, porous ceramic

## Abstract

The process of anaerobic digestion used for methane production can be enhanced by dosing various additive materials. The effects of these materials are dependent on various factors, including the processed substrate, process conditions, and the type and amount of the additive material. As part of the study, three different materials—iron powder, lime, and milled porous ceramic—were added to the 30-day anaerobic digestion of the brewer’s spent grain to improve its performance. Different doses ranging from 0.2 to 2.3 g_TS_ × L^−1^ were tested, and methane production kinetics were determined using the first-order model. The results showed that the methane yield ranged from 281.4 ± 8.0 to 326.1 ± 9.3 mL × g_VS_^−1^, while substrate biodegradation ranged from 56.0 ± 1.6 to 68.1 ± 0.7%. The addition of lime reduced the methane yield at almost all doses by −6.7% to −3.3%, while the addition of iron powder increased the methane yield from 0.8% to 9.8%. The addition of ceramic powder resulted in a methane yield change ranging from −2.6% to 4.6%. These findings suggest that the use of additive materials should be approached with caution, as even slight changes in the amount used can impact methane production.

## 1. Introduction

Brewer’s spent grain (BSG) is a byproduct of beer production. Worldwide spent grain production is around 38.6 × 10^6^ Mg × year^−1^ [1]. Projections indicate further growth in beer production, which will increase the amount of BSG waste generated in the production process [2]. According to Mussatto et al. [3], each hectoliter of beer generates about 22 kg of BSG waste [3]. In Poland, about 889 thousand tons of waste from the brewing industry was generated in 2019, of which BSG accounted for about 85% of the waste mass. BSG waste poses some problems due to its high water content, which is typically around 80%. Brewer’s spent grain is an interesting substrate for use in many industries due to its composition [4]; BSG is rich in protein, fiber, amino acids, and vitamins [5]. However, the use of this material is limited by its high water content. Water affects the acceleration of the BSG rotting process by which this material can become unsafe for use [3]. Due to the high moisture content, high chemical energy content, and relatively easy degradability in anaerobic conditions, BSG can be used as an alternative to commonly used substrates in the methane fermentation process [5]. The biggest advantage of using BSG is its hydration and high-quality composition; of course, these properties are maintained when using fresh BSG directly or stored for up to 3 days at most. In the methane fermentation process, it is important to maintain a high-quality substrate to ensure high and stable biogas production [2,3,6].

The current geopolitical situation is forcing a reconstruction of the existing model of global energy production. Conventional fuels are being replaced by alternative energy sources over time, but their share is still low [7]. The main reason for the slow energy transition is the failure to adapt the existing energy infrastructure to the use of alternative fuels. Currently, in Poland alone, more than half of the population has a gas grid connection; in contrast, industry consumes 40% of the gas from the total stream used in Poland [8]. Biogas is an interesting energy source that has the potential to replace natural gas. Currently, biomethane, or biogas after a purification process, is of great interest. Biogas purification methods increase the share of methane in the gas stream but do not increase its production in the methane fermentation process itself [9]. Optimizing the methane fermentation process to increase the proportion of methane in biogas is sought because the possibility of producing a high-quality fuel similar in quality to natural gas may allow biogas to be distributed with existing infrastructure [10].

Biogas consists of a mixture of different gases, of which about 1/3 is carbon dioxide (CO_2_) and 2/3 methane (CH_4_), as well as water vapor and trace gases [11]. The calorific value of pure methane, which is 35 MJ × m^−3^, is an important measure of its energy potential. In the case of biogas, the energy value can vary depending on the proportion of gases that make up its mixture, especially the proportion of methane. The average calorific value of biogas is about 21 MJ × m^−3^, oscillating between 16 and 23 MJ × m^−3^. If biogas is purified of carbon dioxide, its calorific value can increase to about 35 MJ × m^−3^; this is important because the elimination of CO_2_ increases the energy value of biogas. The energy value of 1 m^3^ of purified biogas is comparable to the energy contained in 0.93 m^3^ of natural gas, 1 dm^3^ of diesel fuel, or 1.25 kg of coal. It can also be said that the energy contained in 1 m^3^ of biogas is equivalent to about 9.4 kWh of electricity [12]. 

From the point of view of a biogas plant operator, the methane content is of paramount importance, as it represents the percentage of methane in the gas mixture that is responsible for energy recovery. The factors affecting the methane content include substrate moisture content, fermentation temperature, hydraulic retention time, substrate preparation, and the degree of its decomposition [12,13]. The quality of substrates has a major impact on the quantity and quality of biogas produced, so it may be necessary to carry out substrate pretreatment. As a general rule, the substrate must be of the highest possible quality. Methane yield is particularly dependent on the composition of the substrate, including the presence of fats, proteins, and carbohydrates. It is worth noting that methane yield decreases in the order of fats, proteins, and carbohydrates [14]. 

The methane fermentation process comprises four phases, each involving different microorganisms and biochemical reactions. The first phase is hydrolysis, in which polymeric organic compounds (fats, proteins, and carbohydrates) are broken down into simpler forms by hydrolytic bacteria. Proteins, fats, lipids, and carbohydrates are converted into amino acids, fatty acids, and simple sugars, respectively. The second phase is acidogenesis, in which the products of hydrolysis are converted by acid-forming bacteria into organic acids, alcohols, amines, and other compounds. The third phase is acetogenesis, in which the products of acidogenesis are converted by acetic bacteria into acetates and carbon dioxide. The fourth and final phase is methanogenesis, in which acetate and alcohol products are converted into methane and carbon dioxide by methanogenic microorganisms [11,15].

In recent years, in addition to optimizing and monitoring the basic parameters of methane fermentation, there has been interest in various additives to enhance biogas production and process stability. One way to increase the efficiency of biogas production is to place porous materials in the digester to improve the adhesion surface of microorganisms and thus increase their population. The most commonly used materials added to the methane fermentation process are activated carbon, biocarbon, zeolite, mineral wool, or porous ceramics [16]. The use of porous materials like natural or modified zeolites results in more efficient removal of organic matter and prevents the leaching of microorganisms. Microorganisms in porous materials find space for their colonies to grow. According to Khor et al. [17], zeolite is an excellent ion exchanger that removes ammonium ions with up to 95% efficiency. Properties of zeolite, such as a highly developed porous structure, minimal pore diameter, and low bulk density, make this material an interesting improver of methane fermentation. It has been shown that clay materials affect the microbiologic and enzymatic transformation of substances inhibiting anaerobic decomposition processes [18]. 

Another method of methane fermentation enhancement is the addition of trace elements. Trace elements are crucial for the proper functioning of enzyme complexes in all organisms. In anaerobic digesters, the control and addition of trace elements are essential for ensuring stable and efficient methane production processes. On the other hand, deficits in trace elements can cause imbalances in the process. Unfortunately, the metabolic mechanisms and adaptation of affected microbial communities to such deficits are not yet fully comprehended [16,19,20]. Many researchers have indicated that trace elements play an important role in the growth of methanogens and the formation of methane. In addition, supplementing with alkaline metals can effectively reduce the acidification caused by the quick breakdown of easily degradable substrates. This acidification occurs due to the accumulation of volatile fatty acids, which are produced by acidogenic microorganisms at a rate that exceeds the processing capabilities of methanogenic microorganisms [21]. 

One of the important trace elements in methane fermentation is iron. It is believed that in microbial metabolism, iron can act as an electron donor and facilitate the functioning of important enzymes during the acetogenesis process shifting propionate concentration to acetate that is subsequently used for methane production by methanogenic bacteria [22]. Furthermore, iron can alleviate low pH by increasing the metabolic efficiency of microorganisms and reducing the oxidation–reduction potential [23]. 

Lime is another material that can be used for methane fermentation improvement or substrate preparation and plays an important role in the decomposition of lignin. Anaerobic microorganisms have a problem biodegrading lignin due to its crystalline structure and accessible surface area, making it difficult for the enzyme to reach the reaction site [24]. To improve the efficiency of the process, lime is added to the substrate before methane fermentation to depolymerize the biomass, leading to the formation of simpler compounds that are more accessible to microorganisms [25]. Various lime types can also be added directly to the anaerobic reactor to increase buffer capacity and stabilize the desired pH [26]. 

Recently, several studies tested the effectiveness of biochar (BC) in enhancing the methane fermentation of a brewery’s spent grain [2,27,28]. Biochar is produced by heating biomass at temperatures over 300 °C in the absence of oxygen. Biochar has a porous structure and high specific surface area with many functional groups, making it an ideal habitat for microorganisms and capable of adsorbing pollutants (e.g., excess ammonia, toxins, and heavy metals). Biochar’s functional groups and ions (e.g., Na^+^, K^+^, Ca^2+^, and Mg^2+^) increase the buffer capacity of the digester, stabilizing its performance, while electrical conductivity increases and promotes direct interspecies electron transfer (DIET) [29,30]. The outcomes of using biochar for the anaerobic digestion (AD) of BSG differ among the research, and in some cases, excessive amounts of biochar significantly reduced methane yield. However, a proper amount of biochar increased the methane/biogas yield by 10% [27], by 26.6% [28], or even by 145% [2]. 

To ensure the optimal performance of a biogas plant, it is crucial to optimize the methane fermentation process. Where additive materials are concerned, it is important to use the right dose in specific situations to obtain the desired effect. For that reason, in this study, we tested the effects of three commonly available material additives (porous ceramics, iron powder, and lime) on the anaerobic digestion of brewer’s spent grain. Though the chosen materials have been tested individually on various substrates by other researchers, the authors have observed a lack of studies examining the effects of these materials on the mesophilic methane fermentation of brewer’s spent grain. To address this, this study tested different doses of three materials and their mixtures to determine their impact on methane production and kinetics. 

## 2. Materials and Methods

### 2.1. Materials

Digestate, used as inoculum for the AD experiment, was collected from an agricultural biogas plant (Bio-Wat Sp. Z o. o., Świdnica, Poland). The biogas plant had a total electrical power of 1 MW and was fed with maize silage and other unspecified seasonal agricultural substrates. The digestate was collected from a post-fermentation chamber. On the same day, the digestate was transported to the laboratory and strained through a piece of fabric to remove large solids particles. Afterward, the liquid digestate was poured into 5 dm^3^ containers (not completely tight) and placed in the laboratory incubator (POL-EKO-APARATURA, model ST 3 COMF, Wodzisław Śląski, Poland) at 4 °C. The liquid digestate was heated to room temperature before use in the experiment.

The main substrate, the brewer’s spent grains was collected from a small-scale beer production installation (Wroclaw University of Environmental and Life Sciences, Wrocław, Poland). Wet BSG was dried at 80 °C using a laboratory dryer (WAMED, model KBC-65W, Warsaw, Poland) and stored in plastics bags that were placed into the laboratory freezer (Electrolux, model EC5231AOW, Jászberény, Hungary) at −31 °C. Before use in the AD experiment, the BSG was defrosted and heated to room temperature. 

The additive materials used in the methane fermentation experiment were milled porous ceramics (Cf), iron powder (Fe), and lime (L) (calcium hydroxide (Ca(OH)_2_)). Porous ceramics were made from ceramic rings used as ceramic cartridges for aquariums (NCR-0.5, Aqua Nova). The ceramic rings were crushed into powder. Fe and Ca(OH)_2_ were purchased from (Warchem Sp. Z o.o., Zakręt, Poland), as high-purity chemicals. 

### 2.2. Methods

#### 2.2.1. Materials Characterization

The materials used and generated in the study (process residues), with some exceptions, were analyzed for total solids (TS), volatile solids (VS), ash content (AC), pH, electrical conductivity (EC), elemental contents (C, H, N, S, O), and trace elements content (Fe, Co, Mo, Se, W, Cu, Zn, Mn). The TS, VS, and AC were determined using a laboratory dryer (WAMED, model KBC-65W, Warsaw, Poland) and muffle furnace (SNOL, model 8.1/1100, Utena, Lithuania) according to the PN-EN 14346:2011 and 15169:2011 standards [31,32]. The pH and EC were measured using a ph/EC meter (Elmetron, model CPC-411, Zabrze, Poland). For the dry materials, the pH and EC measurements were made in material-to-distilled water solutions of 1:10 by mass, while for the liquid materials, direct measurement was made. The elemental content was determined using an elemental analyzer (PerkinElmer, 2400 CHNS/O Series II, Waltham, MA, USA). The trace element content was determined using an inductively coupled plasma atomic emission spectrometry (Thermo Scientific, model 7400, Waltham, MA, USA) according to the PN-EN ISO 118852009 standard [33]. 

#### 2.2.2. Anaerobic Digestion Experiment

The anaerobic digestion experiment was performed using the automatic methane potential test system (BPC Instruments AB, model AMPTS^®^ II, Lund, Sweden). The system consists of 15 batch reactors of 0.5 L volume each, placed in a water bath set at 37 °C. The reactor’s content was mixed by an agitation system. The experiment was performed in three trials, each taking 30 days. In each trial, another additive material was tested. Each trial consisted of 1 reactor filled with inoculum, 2 reactors filled with inoculum and BSG, and 12 reactors filled with inoculum, BSG, and the additive material. A schematic presentation of the reactor’s setup in the specific trials is presented in Figure A1. The 12 reactors filled with additive material contained 6 variants of different additive material loadings. The amount of specific additive material was added to obtain an additive material-to-BSG share of 1, 3, 5, 7, 9, and 12% by total solids. The specific mass of inoculum, substrate, and additive material placed into the reactor for specific variants is presented in Table 1. Each reactor was filled with 150 g of wet inoculum, 4.71 g of dry BSG, and an appropriate amount of additive material. The substrate-to-inoculum ratio (SIR) by volatile solids was kept at 0.61. At the beginning and the end of the process, the pH and EC were measured. 

It turned out that, at some doses, the ceramic and iron powders had a positive effect on the anaerobic digestion of BSG (the higher methane production concerning control). For that reason, an additional (fourth) trial was performed. The purpose was to determine the possible synergistic or antagonistic interactions that may occur between additive materials. The experiment was performed in the same manner, but ceramic and iron powders were mixed and added to reactors in amounts to obtain 3, 6, and 9% of additive materials concerning the substrate. The experiment consisted of:Three reactors filled with 150 g of wet inoculum (variant name—D);Three reactors filled with 150 g of wet inoculum and 4.71 g of dry BSG (variant name—Fe0&Cf0);Three reactors filled with 150 g of wet inoculum, 4.71 g of dry BSG, 0.09 g of iron powder, and 0.09 g of ceramic powder (variant name—Fe3&Cf3);Three reactors filled with 150 g of wet inoculum, 4.71 g of dry BSG, 0.145 g of iron powder, and 0.145 g of ceramic powder (variant name—Fe6&Cf6);Three reactors filled with 150 g of wet inoculum, 4.71 g of dry BSG, 0.26 g of iron powder, and 0.26 g of ceramic powder (variant name—Fe9&Cf9).

The schematic presentation of the experiment setup is presented in Appendix B in Figure A1. The raw data of biomethane production of all trials are given in the Appendix A.

#### 2.2.3. Kinetics Parameter Determination

The cumulative methane production results obtained from the anaerobic digestion experiment were subjected to the determination of the kinetic parameters. Kinetics parameters allow for mathematical descriptions of the curves and an easier comparison between obtained results. The first-order model was used. The raw data of the cumulative biomethane production was subjected to the nonlinear regression analysis to fit experimental data to Equation (1). During regression, the raw experimental data (*y_t_* and *t*) were fitted to Equation (1) using the least square methods to determine kinetics parameters (*y_m_* and *k*). The calculations were performed using Statistica software (TIBCO, version 13.0, Palo Alto, CA, USA). Next, using determined values of *y_m_* and *k*, the additional parameters were determined using Equations (2) and (3).
(1)$$y_{t}=y_{m}\times \left(1-EXP\left(-k\times t\right)\right)$$
(2)$$r=k\times y_{m}$$
(3)$$t_{1/2}=\frac{ln2}{k}$$
where 𝑦_𝑡_—experimental biomethane yield obtained after time t, mL × g_VS_^−1^; 𝑦_𝑚_—estimated maximal methane yield, mL × g_VS_^−1^; 𝑘—methane production constant, d^−1^; 𝑡—process time, d; *r*—biomethane production rate, mL × g_VS_^−1^ × d^−1^; and $t_{1/2}$—half-life time of maximal methane production, d.

#### 2.2.4. Substrate Conversion Efficiency

The theoretical maximal methane potential (*y_t_*) of the processed brewery spent grain was calculated using Boyle’s stoichiometric formula presented in Equation (4). Afterward, to determine the efficiency of the BSG’s conversion into methane-obtained biodegradability (*BD*) was calculated according to Equation (5). To check the organic matter utilization, volatile solids removal (*VS_r_*) according to Equation (6) was also calculated.
(4)$$C_{a}H_{b}O_{c}N_{d}S_{e}+\left(a-\frac{b}{4}-\frac{c}{2}+\frac{3d}{4}+\frac{e}{2}\right)H_{2}O\to \left(\frac{a}{2}+\frac{b}{8}-\frac{c}{4}-\frac{3d}{8}-\frac{e}{4}\right)CH_{4}+\left(\frac{a}{2}-\frac{b}{8}+\frac{c}{4}+\frac{3d}{8}+\frac{e}{4}\right)CO_{2}+dNH_{3}+eH_{2}S$$
(5)$$BD=\frac{y_{exp}}{y_{t}}\times 100$$
(6)$$VS_{r}=\left(\frac{VS_{added}-VS_{final}}{VS_{added}}\right)\times 100$$
where *C_a_H_b_O_c_N_d_S_e_* is an elemental composition of processed brewery spent grain; *a*, *b*, *c*, *d*, *e* is a molar % shares of those elements contained in the volatile solids; *y_exp_*—experimental methane yield; mL × g_VS_^−1^; and *y_t_*—theoretical maximal biochemical methane potential, mL × g_VS_^−1^; $VS_{r}$—volatile solids removal, %; $VS_{added}$—volatile solids at the start of the AD process, g; $VS_{final}$—volatile solids after the AD process, g.

#### 2.2.5. Statistical Analysis

The results of kinetics parameters and substrate conversion efficiency were subjected to an ANOVA at *p* < 0.05 to check if additive materials affected these parameters. When statistically significant differences occurred, a post hoc Tukey’s test was performed to check between which groups those differences occurred. The analyses were performed using Statistica software (TIBCO, version 13.0, Palo Alto, CA, USA).

## 3. Results and Discussion

### 3.1. Materials

The liquid digestate used as an inoculum in the anaerobic digestion process was characterized by a total solids content of 6.8% and volatile solids content of 46.9%. The fresh brewery spent grain obtained after the beer production process had a TS of 20.4% and VS of 96.2%. The elemental analysis revealed that the dry inoculum consisted of 22.5% of C, 2.4% of H, 3.3% of N, 2% of S, 12.3% of O, and 57.5% of ash, while the dry BSG consisted of 48.6% of C, 7.0% of H, 4.4% of N, 2.0% of S, 35% of O, and 3.1% of ash. As a result, the theoretical maximal methane potential (*y_t_*) calculated using Equation (4) for inoculum was 494 mL × g_VS_^−1,^ while for the BSG was 510 mL × g_VS_^−1^. The inoculum pH and EC were 7.75 and 32.4 µS × cm^−1^, respectively, while the BSG pH and EC were 6.4 and 718 µS × cm^−1^, respectively. 

The above-mentioned parameters were not determined for the additive materials (iron particles—Fe, lime—L, and milled ceramic filter—Cf) due to their specific characteristics. All the materials were dry (TS~100%) and were supposed to have no organic matter (VS~0%). Fe and Cf are inert materials that do not change pH, while L is highly alkaline. Fe supposes to increase EC while having a low/non-effect on the pores’ volume and pH, Cf supposes to affect only pore volume, and L supposes to affect mainly pH and EC due to an increase of soluble ions. 

### 3.2. Anaerobic Digestion

In Figure 1, Figure 2 and Figure 3 the results of the additive materials dosed to the AD process are presented. The addition of iron particles did not change the initial pH and EC. The pH in the reactors differed from 7.72 to 7.75, while the EC differed from 31.08 to 31.83 µS × cm^−1^ without specific trends. During AD with Fe, after 30 days of the process, the control reactor produced 316.2 ± 10.8 mL × g_VS_^−1^ of methane. Fe supplementation in shares of 1, 3, 5, 7, 9, and 12% resulted in a methane production of 326.1 ± 9.3, 347.2 ± 3.4, 309 ± 10, 317.2 ± 3.5, 313.8 ± 6.7, and 314.5 ± 3.4 mL × g_VS_^−1^, respectively (Figure 1). As a result, Fe supplementations of up to 3% of the BSG’s total solids increased methane production, while higher Fe doses slightly decreased it. The highest increase in methane production was observed for Fe3 (9.8%), while the lowest increase was for Fe9 (0.8%). Despite the differences observed, the post hoc Tukey’s test indicated that there were no statistically significant variations in methane yields among the tested Fe doses.

Although the iron powder did not do its job and no increase in electrical conductivity was observed, many studies show that iron supplementation can enhance methane production, substrate conversion, process stability, and increase the reduction of H_2_S [34]. Iron is one of many essential trace metals needed during the AD process and, similar to other heavy metals, constitutes part of enzymes that drive various AD reactions. Overload, as well as depletion of specific trace metals, will result in the inhibition of AD microorganisms. The effect does not only depend on total metal concentration but also its chemical forms, pH, and redox potential [34,35]. Each group of AD microorganisms has its optimal trace element concentration, and it is believed that acidogens are more resistant to heavy metals overloading than methanogens [35]. For this reason, different results regarding iron supplementation among studies can be found. In the work of Andriamanohiarisoamanana et al. [36], iron powder (85% Fe_3_O_4_) was added to dairy manure at a concentration between 2 and 20 g × L^−1^. The iron powder did not change the methane yield significantly, but the hydrolysis constant rate increased by ~100%, the lag phase was reduced to half, and the reduction of ~99% of H_2_S was obtained concerning the reactor without Fe_3_O_4_ supplementation [36]. On the contrary, Liu et al. [37] tested different types of zero-valent iron in the form of iron powder, clean scrap, and rusty scrap at doses of 1–4 g × L^−1^. As a result, the methane yield increased from 248 to 300 mL × g_VS_^−1^ (an increase of 21%), while hydrolysis did not change (0.083 d^−1^) [37]. Cao et al. [38] tested the effects of zero valency iron powder (Fe^0^) at a dosage of 30 mg × g_VS_^−1^ on the AD of sewage sludge. Fe^0^ addition resulted in a small reduction in the diversity of the archaeal community that decreased from 1534 OTUs to 1493 OTUs, and a significant increase in the content of hydrogenotrophic methanogens (by 15.4%) was observed. Increased abundance of hydrogenotrophic methanogens helped to reduce hydrogen partial pressure, thereby increasing acetic acid content and methane production by 18.2% [38]. It is worth noting that hydrogenotrophic methanogens are a group of slow-growing microorganisms that convert dissolved CO_2_ and H_2_ (CO_2_ + 4H_2_→CH_4_ + 2H_2_O), which during normal AD are responsible for around 1/3 of total methane production [39]. Meng et al. [22] studied the effects of Fe^0^ addition to the acidogenic reactor that processed artificial wastewater. The results show that Fe^0^ powder enhances the conversion of propionate to acetate, raising acetate production and chemical oxygen demand (COD) removal [22]. 

In the case of AD supplemented with lime (Figure 2), the control reactor produced 301.7 ± 16.7 mL × g_VS_^−1^ of methane. The control reactors’ pH was 7.84, and lime addition slightly increased its value up to 7.86. There were no significant changes in EC values. The methane yield after 30 days of the process for reactors with lime added in 1, 3, 5, 7, 9, and 12% shares were 281.4 ± 8.0, 286.7 ± 4.3, 291.8 ± 3.7, 285.5 ± 8.5, 302.2 ± 11.8, and 287.2 ± 7.9 mL × g_VS_^−1^, respectively (Figure 2). Only reactor L9 had a similar methane yield as the control, while other reactors showed a methane yield decrease in the range from −6.7% to −3.3%, suggesting that for tested materials, lime addition had a negative effect. However, the post hoc Tukey’s test showed no statistically significant differences in methane yield between tested lime doses. Interestingly, for unknown reasons, between days 12 and 16, a lag phase in methane production occurred. Such phenomena were not observed for experiments with Fe and Cf, though the same BSG and inoculum were used.

For the performed experiment, lime turned out to not improve methane production. The lack of process improvement was probably because the pH of the process was already in the optimal range. Although it was expected that the alkaline properties of lime would increase pH and improve process stability, the tested organic loading (a SIR by VS of 0.61) did not lead to digester overloading, and the drop in pH did not harm or inhibit the methanogenic microorganisms. For example, Zhang et al. [40] studied lime loadings of 0, 0.015, 0.03, and 0.05 g_Ca(OH)2_ × g_dry biomass_^−1^ on the AD of smooth cordgrass and observed biogas production decreasing by 7.1%, 20%, and 75.7%, respectively. The decrease in biogas production was probably due to a too-high initial pH (11.1–12.9), and though the pH stabilized quickly at the optimum range of around 6.5–7.5, the methanogenic bacteria activity was disturbed [40]. What is more, other microorganisms were not inhibited since volatile fatty acids (VFAs) were still produced and accumulated [40]. In turn, Zhang et al. [41] studied the impact of using lime mud, a byproduct of the papermaking process, as a buffering agent and inorganic nutrient on the stability of the mesophilic AD of food waste. The lime mud was primarily comprised of CaCO_3_ and CaO, and the experiment involved lime mud doses of 0, 2, 6, 10, and 14 g × L^−1^. An increase in lime dosage up to 10 g × L^−1^ significantly improved methane production, while a lack of lime (control) resulted in almost complete inhibition. On the other hand, increasing the mud load over >10 g × L^−1^ started to reduce methane production [41], showing that a proper amount of lime can stabilize the process, while too much can decrease its efficiency. As a result, a proper amount of lime added during anaerobic digestion can help to slower volatile fatty acids release, maintaining a pH level that is beneficial for the survival of methanogenic bacteria [41]. The optimal pH range for a one-stage AD process is generally between 7 and 8 [42], but the range differs for each type of microorganism. Fermentative bacteria can thrive at a pH of 4–8.5 with an optimum of 5–6, while methane-producing archaea can survive at 5.5–8.0, with an optimal range of 6.5–8.0 [39]. For this reason, maintaining the appropriate levels of acidity and alkalinity is crucial in the anaerobic digestion process to ensure efficient methanogenic activity and metabolism pathways. Deviations in pH, VFAs, or alkalinity levels can obstruct microbial growth, leading to the inhibition of CH_4_ production. Careful monitoring and balancing of these factors (e.g., by adding lime) are essential to achieve high biogas/biomethane production [26,40,41,43].

During the trial with porous ceramic powder supplementation, the methane yield from the control was 311.8 ± 3.1%. The addition of ceramic did not change both pH and EC significantly, which varied from 7.77 to 7.80 and from 41 to 45 µS × cm^−1^, respectively. The methane yield from the reactors Cf0–12 was 308.0 ± 4.0, 310.9 ± 0.3, 308.0 ± 3.2, 326.1 ± 31.3, 303.7 ± 1.4, and 311.9 ± 2.7 mL × g_VS_^−1^, respectively (Figure 3). As a result, the change in methane production considering the control was −1.0%, −0.3%, −1.2%, 4.6%, −2.6%, and 0.01%, respectively. The highest increase in methane production was obtained for reactor Cf7 (4.6%), while the highest decrease was for reactor Cf9 (−2.6%). Interestingly, the post hoc Tukey’s test showed no statistically significant differences between all the variants containing ceramic powder and the control without ceramic powder. However, statistically significant differences occurred between variants Cf3 and Cf5 that contained a 3 and 5% share of lime, respectively. Contrary to trials with iron powder and lime, the course of cumulative methane production for ceramic powder has no disturbance, suggesting a lack of impact on process stability.

Porous materials are widely used in the anaerobic digestion process as a support medium for bacteria colonization by increasing the available surface. When fixed beds are used, porous materials immobilize the microorganisms’ biomass, thus, increasing AD performance. The most common porous materials are natural zeolites [44] and carbonaceous materials, e.g., biochar [45]. Other porous materials used for bacterial adhesion and, thus, increasing the microorganisms’ population are bentonite, mineral wool, polyurethane, polyacrylate, polyethylene, and straw [46]. Regardless of the porous material, the pores must be large enough for methanogenic bacteria populations to colonize. Each bacterium is about 1 µm in size, and for that reason, the pore size and distribution, and the way of its usage (as an additive or as a fixed bed) affect the microbial community and AD performance [47]. In addition to the porous structure, other specific properties of used materials (e.g., surface functional groups, existence of metals, specific surface area ion exchange capacity, etc.) can affect the AD process, and the final effects will depend on the synergetic/antagonistic interactions [44,45,48]. For example, Montalvo et al. [49] studied the effects of particle size and doses of zeolite and sand addition on the AD of synthetic and piggery wastes. Doses of 0.05 to 0.40 g of zeolite per g of volatile suspended solids (VSS) were used and the mechanisms of AD enhancement depended on a processed substrate. For piggery waste, the methane yield increase was related to microorganisms’ immobilization on zeolite, while for synthetic waste, the methane increase was related to microorganism immobilization and concentration reduction of toxic nitrogen by zeolite [49]. The results showed that 0.10 g_zeolite_ × g_VSS_^−1^ was the most beneficial for total chemical oxygen demand (TCOD) decrease and methane yield increase, while higher doses up to 0.30 g_zeolite_ × g_VSS_^−1^ resulted in less process performance improvement. It is noteworthy that the addition of 0.40 g_zeolite_ × g_VSS_^−1^ resulted in a decline in the process performance compared to the control group without any supplementation. Similar results were obtained by Shi et al. [48], who investigated the impact of different biochar doses on suppressed mesophilic anaerobic digestion of oily sludge (OS). Biochar doses of 0.6, 1.2, 2.4, and 4.8 g_biochar_ × g_VS of OS_^−1^ were used. All doses up to 2.4 g_biochar_ × g_VS of OS_^−1^ resulted in process performance improvement concerning the control, while 4.8 g_biochar_ × g_VS of OS_^−1^ turned out to be excessive, resulting in negative effects with methanogenic efficiency, an extended lag phase, and decreasing total methane yield [48]. This shows that the proper amount of porous material can improve the AD process, while too much can result in its disturbance. When using a high dose of porous material, it can reduce the amount of free water available. This can affect the transportation of nutrients and metabolites near the porous material particles and associated microorganisms. Consequently, using a large amount of porous material can increase the medium’s apparent viscosity, which can hinder the mass transfer between the substrate and microorganisms responsible for the process, ultimately slowing down the anaerobic digestion process [48,49]. Considering the above, it can be concluded that in the performed research (Figure 3), the amount of used ceramic powder could be too small to significantly improve the microorganism’s community and too small to significantly interrupt the mass transfer between the substrate (dissolved organic matter from the BSG) and the microorganisms, since the difference in obtained methane yield did not differ significantly.

Figure 4 presents the results of an additional experiment whose purpose was to check the possible occurrence of synergistic interactions between the additive materials when used at the same time. For the experiment, ceramic and iron powders were chosen because some of the doses have been shown to have positive effects on methane production (Figure 1 and Figure 3), while lime reduced methane yield at all tested doses (Figure 2). The control reactors without ceramic and iron powders after 30 days produced 305.3 ± 2.7 mL × g_VS_^−1^. The variants with 3, 6, and 9% of additive materials produced 305.2 ± 8.2, 298.1 ± 0.9, and 299.1 ± 6.2 mL × g_VS_^−1^, respectively (Figure 4). The differences turned out to be statistically insignificant, showing a lack of methane yield improvement and a lack of synergistic/antagonistic interactions between the studied materials. The lack of methane yield improvement may result from the fact that the experiment was performed at a suitable substrate-to-inoculum ratio (SIR) for microorganisms, thus, both over- or under-loading of the process was avoided [50,51], or from the fact that added materials were too low to affect an already optimized process. Thus, additional ceramic pores were not used for microorganisms’ growth (there was probably enough in place before ceramic was used), while the iron was probably not used to enhance the conversion of propionate to acetate (the process was not overloaded and there was no excessive accumulation of propionate).

To facilitate a more accurate comparison of obtained results with those of other researchers, we have tabulated our and their experimental setups and primary findings in Table 2. As far as we know, no other research has investigated the impact of iron, porous ceramic, and lime powder individually and in mixtures on the anaerobic digestion of brewer’s spent grain. For this reason, we have tabulated research where similar substrates or additive materials with similar properties were used. As can be seen in Table 2, different results can be obtained using similar additive materials but at different conditions (other types of reactor, substrate and inoculum properties, SIR ratio, process temperature, and time, etc.). In the case of iron powder or Fe-containing materials (e.g., rusty scrap) in all comprised research, the Fe addition resulted in process performance improvement or did not affect it negatively, though some side effects, such as changes in the microbial community, can occur (Table 2). On the other hand, lime effects are more complex and depend more on specific scenarios (type of processed substrate and process conditions). In the case of reactors operated at optimal conditions (this study, SIR by VS 0.61) or those fed with not easy/fast degradable substrate [40], lime addition results in methane yield, and its production kinetics decrease (Table 2). In cases where easily degradable substrates (e.g., food waste) are being processed or when the digester is overloaded with organic matter, adding lime stabilizes the process by increasing buffer capacity and maintaining proper pH levels. Lime, which is a source of trace elements (such as lime mud from papermaking), also helps microorganisms maintain good health and become more resistant to unfavorable conditions (Table 2). Even more complex results in AD performance can be observed when porous materials, e.g., porous ceramic, zeolite, bentonite, biochar, and their modifications, are used (Table 2). Although porous material naturally creates more surfaces for microorganisms to grow and adsorb contaminants, leading to improved process efficiency, the modified materials can have an impact on the process at multiple levels. Zeolites, Fe-metal organic frameworks, biochars, etc., apart from the relatively high surface area and pore abundance, can improve buffer capacity through the presence of functional groups and ions, as well as increase electrical conductivity. Moreover, biochars and other conductive carbon-based materials are considered to promote direct interspecies electron transfer (DIET) between syntrophic bacteria and methanogens, accelerating substrate degradation, lowering volatile fatty acids concentration, and increasing methane production [29]. Although a considerable amount of research is available, it remains uncertain how particular properties interact under specific anaerobic digestion conditions. Therefore, in certain instances, additive materials can boost process efficiency, while in others, they may diminish it (Table 2). In the performed research, the porous material (Figure 3), as well as its mixture with conductive material (iron powder) (Figure 4), did not significantly change the process performance. Nevertheless, there is evidence that porous, conductive carbon-based materials (biochars) can improve methane yield from BSG by 1.8%, 10%, and 3.1% at a SIR of 0.5, 1.0, and 2.0, respectively [27], or can alleviate digester acidification, increasing methane yield by 26.6% [28], or increase biogas yield by 145% and biogas production constant (k) by 30% [2]. However, there are also pieces of evidence that the same biochar at different doses/conditions may disturb the process [2,28]. 

### 3.3. Kinetics Parameters and Substrate Conversion Efficiency

To study the effects of additive materials on the methane production process throughout the range and not just focusing on the final methane yield after 30 days, kinetics parameters are determined and summarized in Table 2 alongside substrate conversion efficiency. 

The estimated maximal methane yield (𝑦_𝑚ax_) in the trial with Fe varied from 383.6 ± 2.6 mL × g_VS_^−1^ for the control (Fe0) to 403 ± 9.2 mL × g_VS_^−1^ for Fe3. At the same time, no significant differences were observed for the methane production constant (k) that varied in a much narrower range from 0.07 to 0.08 d^−1^. The biomethane production rate (r) varied from 26.4 ± 1.5 to 31.7 ± 1.5 mL × (g_VS_ × d)^−1^ and was the lowest for Fe0 and the highest for Fe3. In the case of a half-life time of maximal methane production ($t_{1/2}$), half of the maximal estimated value was obtained the fastest by reactors Fe5 (8.6 ± 0.5 d), while in the control after 10.1 ± 0.5 d (Table 2).

For the trial with lime, the 𝑦𝑚 varied from 305.0 ± 2.0 to 329.5 ± 13, and k varied from 0.10 to 0.12 d^−1^ without a specific trend. Though there was no specific trend for 𝑦𝑚 and k, the methane production rate decreased with increasing lime load. The r decreased from 40.1 ± 5.2 mL × (g_VS_ × d)^−1^ for the control to 32.5 ± 2.7 mL × (g_VS_ × d)^−1^ for L12. In contradiction to Fe addition, lime addition resulted in an extension of time needed to obtain half of the maximal methane production from 5.6 ± 0.6 d (L0) to 6.8 ± 0.6 (L12) (Table 3).

In the case of the AD reactors with porous ceramics, the 𝑦_𝑚ax_ varied from 296.2 ± 2.3 mL × g_VS_^−1^ to 300.7 ± 2.0 mL × g_VS_^−1^, k varied from 0.23 d^−1^ to 0.25 d^−1^, r varied from 68.1 ± 2.4 mL × (g_VS_ × d)^−1^ to 71.7 ± 2.5 mL × (g_VS_ × d)^−1^, and $t_{1/2}$ varied from 2.8 ± 0.1 d to 3.1 ± 0.0 d. The obtained values did not differ significantly (*p* < 0.05) from the control, and the mean values of specific variants were covered by standard deviations of other variants suggesting that the addition of ceramic powder had no significant effect on methane production kinetics (Table 3).

The obtained biodegradability (BD) and volatile solids removal (VSr) slightly differed between trials. The mean BD value from all control reactors was 60.9%, while VSr) was 44.1%. For the trial with Fe, the control reactor obtained a BSG conversion to methane of 62.0 ± 2.1%, while for the L and Cf trials, the control reactors obtained 58.8 ± 3.3% and 61.9 ± 0.7%, respectively while for VSr, these values were 43.0 ± 0.9%, 44.7 ± 0.0%, and 44.7 ± 2.1%, respectively (Table 3). In the case of the Fe trial, the highest BD was obtained for Fe3 (68.1 ± 0.7%), and the highest VSr was obtained for Fe9 (44.3 ± 1.5%). For the lime trials, almost in all cases, the addition of lime resulted in a decreasing BD. The lime addition did not have specific effects on VSr. In the case of the porous ceramics trial, the highest BD was obtained by Cf7 (64.8 ± 6.2%), while the highest VSr was obtained by Cf1 (45.6 ± 1.0%) (Table 3).

### 3.4. Process Residues

The characteristics of the digestate (process residues) are summarized in Table 4 and Table 5. The mean value of the organic matter (VS) in the process residues in the control reactors was 63.2%, while the inorganic matter (AC) consisted of 36.8% of dry matter. The elemental analysis of the dry mass revealed that the process residues from the control reactors consisted of 33.7% of C, 4.5% of H, 3.9% of N, 1.2% of S, and 20.9% of O. As a result of similar methane yield and obtained biodegradation, as well as volatile solids removal (Table 4), the process residues were characterized by similar organic matter content and its composition. 

In the case of the Fe trial, no differences were found between the presented data, suggesting that Fe addition did not affect the process residue quality. A small EC increase was observed between the control Fe0 (33.8 µS × cm^−1^) and others, Fe1–12 (>34.2 µS × cm^−1^) (Table 4); nevertheless, the difference was not statistically significant (*p* <0.05), suggesting that the addition of iron powder does not help to increase electrical conductivity.

For the lime trial, the obtained results of the process residues were similar to those from the Fe trial. Due to the mineral nature of lime, a small decrease in volatile solids and a small increase in ash content with increasing lime content was visible. The VS decreased from 62.4 ± 1.2% for the control (L0) to 61.0 ± 0.1% for L12, while the ash content increased from 37.6 ± 1.2% to 39.0 ± 0.1% for those reactors. Interestingly, no differences in final pH were observed, though it was assumed that lime would affect pH. The addition of ceramic powder did not change significantly analyzed properties, and results were similar to those from the Fe and lime trials. Here, a small decrease in VS and increase in AC was also observed, with an increasing ceramic powder share increase (Table 4). 

Digestate from agricultural substrates is generally considered to have good fertilizer properties [56]. The quality of digestate differs significantly depending on processed substrates, the used technology, and the process operational parameters. Digestate from agricultural biogas plants is typically characterized by a total solids content of 3.2–6.6%, a volatile solids content of 61–76.5%, and a pH of 8.2–9.4 [57]. In the case of the batch reactors where no in-and-out fluent takes place, the final concentration of specific elements (C, H, N, S, O) contained in the residual mass depends on the initial content in the used substrate and inoculum and the produced biogas’ quantity and quality. As a result of organic matter conversion to biogas, volatile solids are reduced and specific elements leave the reactors in the form of biogas. Biogas may consist of 40–75% of CH_4_, 15–60% of CO_2_, 1–5% of H_2_O (as vapor), and other gases like NH_3_, and H_2_S < 1% and specific biogas composition depend on the processed substrate and process operational parameters [58,59]. As a result, the total amount of elements decreases in favor of the relative increase of ash content. The most important elements of digestate as fertilizer are the concentration of elemental carbon and nitrogen. Depending on the processed substrate, these elements varied from 29.1 to 40.9% and from 4.72 to 16.4% (by dry mass), respectively [60]. Due to volatile solids removal, process residues are characterized by higher ash content than processed substrates. In the case of digestate from agricultural plants, the inorganic fraction may consist of 23.5–39% of dry mass of residues [57]. It seems that the addition of tested additive materials, even in high doses of up to 12% of used substrates, did not change process residue quality since most of its properties are in the range of those that can be found in other studies.

The concentration of micro and macro elements depends on the quality of substrates placed into the reactor. The digestate used in the research was characterized by Fe of 4050 ± 600, Co of 1.85 ± 0.35, Mo of 1.15 ± 0.25, Se of <0.5, W of <0.5, Cu of 31 ± 6, Zn of 225 ± 45, and Mn of 110 ± 20, while the BSG was characterized by Fe of 570 ± 60, Co of <0.20, Mo of 0.68 ± 0.14, Se of <0.4, W of <0.5, Cu of 13 ± 3, Zn of 63 ± 13, and Mn of 26 ± 5 mg × kg_TS_^−1^.

The trace elements contained in process residues are summarized in Table 5. As could be expected, the addition of Fe to the AD process increased its content in process residues. The Fe content increased significantly from 3300 ± 500 mg × kg_TS_^−1^ for Fe0 to 8650 ± 1300 mg × kg_TS_^−1^ for Fe12. Furthermore, other trace element concentrations in the process slurry were similar to those contained in the used digestate, regardless of the trial. The Co varied from 1.5 ± 0.3 to 2.5 ± 0.6 mg × kg_TS_^−1^, Mo varied from 1.2 ± 0.2 to 2.9 ± 0.6 mg × kg_TS_^−1^, Cu varied from 25.5 ± 5.5 to 33.5 ± 6.5 mg × kg_TS_^−1^, Zn varied from 175 ± 35 to 240 ± 50 mg × kg_TS_^−1^, and Mn varied from 87 ± 18 to 104 ± 35 mg × kg_TS_^−1^ (Table 5). In the case of Se and W, the concentrations were below the limit of detection of <5 mg × kg_TS_^−1^. Due to the high deviation in obtained results, there were significant differences between the tested variants, except Fe in the trial with Fe addition. As a result, the tested materials did not affect the quality of process residues.

## 4. Conclusions

In this research, three additive materials were added to a 30-day long, mesophilic anaerobic digestion of brewer’s spent grain operated at optimal conditions (SIR = 0.6). the effects of iron powder, lime, and milled porous ceramic at doses of 0.2, 0.6, 0.95, 1.34, 1.7, and 2.3 g_TS_ × L^−1^ were tested. The impact of lime addition on methane yield was mostly negative, ranging from −6.7% to −3.3%. However, the addition of iron powder showed an increase in methane yield, ranging from 0.8% to 9.8%. The effect of ceramic powder on methane yield was mixed, with changes ranging from −2.6% to 4.6%. Though a different final methane yield was obtained, the kinetics of methane production did not differ significantly. Based on the results of the research, to increase methane yield from mesophilic anaerobic digestion of BSG, iron powder at a dose of 0.6 g × L^−1^, or ceramic powder at a dose of 1.34 g × L^−1^ can be used. Such supplementation may increase methane yield by 9.8 and 4.6%, respectively. However, the research revealed a lack of synergetic or antagonistic effects when iron was mixed with ceramic powder; thus, there is no point in using those additive materials simultaneously in one process. In this study, the addition of additive materials did not have a significant impact on methane production. This could be because the process was already being carried out in optimal conditions, making any further improvements impossible. Therefore, future research should explore the use of additive materials in processes that are not operating at their optimal level. It would be beneficial for the research to include an analysis of the microbial communities and concentrations of volatile fatty acids. This would give a better understanding of how the additive materials impact the AD process. To make the findings more applicable, it is recommended to use continuous-type reactors during the experiment. These reactors generally have higher process kinetics compared to batch reactors, and when operated under similar conditions to industrial plants, the results could be directly implemented.

## Figures and Tables

**Figure 1 materials-16-05245-f001:**
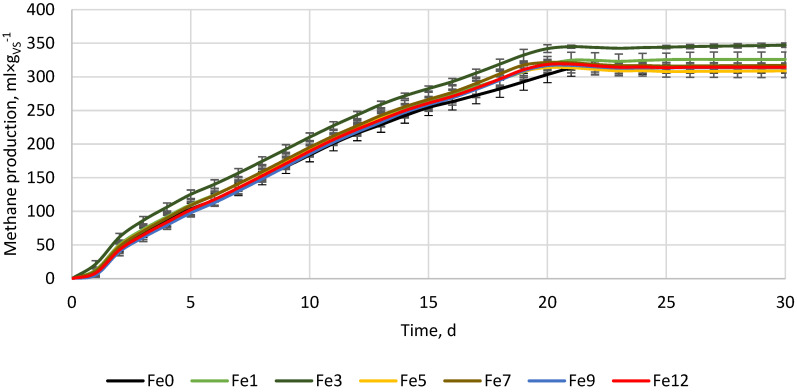
Methane production from the BSG supplemented with iron particles (Fe).

**Figure 2 materials-16-05245-f002:**
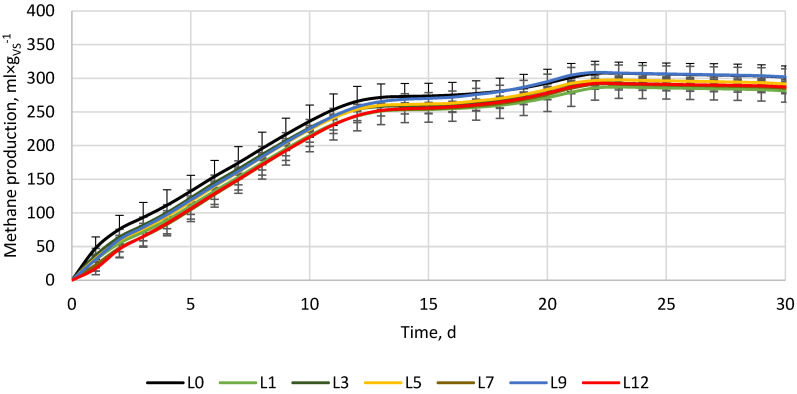
Methane production from the BSG supplemented with lime (Ca(OH)_2_).

**Figure 3 materials-16-05245-f003:**
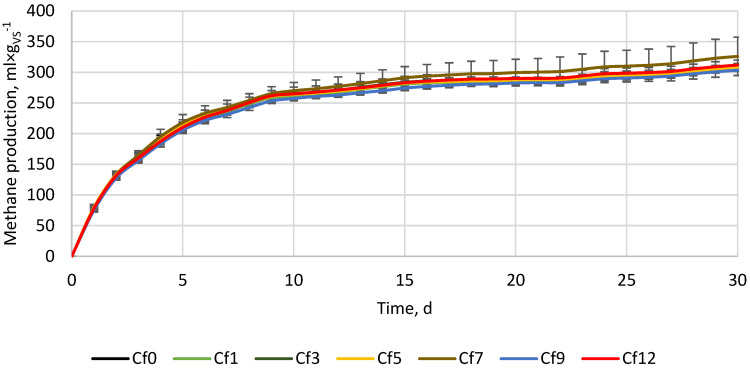
Methane production from the BSG supplemented with milled ceramic powder.

**Figure 4 materials-16-05245-f004:**
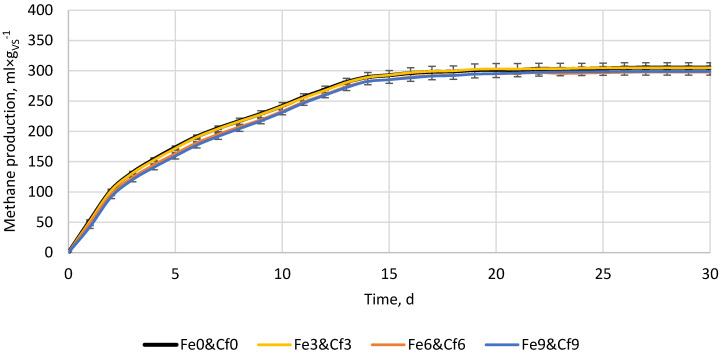
Methane production from the BSG supplemented with different amounts of iron powder and ceramic powder.

**Table 1 materials-16-05245-t001:** Anaerobic digestion experiment matrix and reactor setup.

Variant Name, -	D	Fe/L/Cf0	Fe/L/Cf1	Fe/L/Cf3	Fe/L/Cf5	Fe/L/Cf7	Fe/L/Cf9	Fe/L/Cf12
Substrate, g_wet_ *	0.00	11.7						
Substrate, g_TS_	0.00	3.00						
Substrate, g_VS_	0.00	2.88						
Inoculum, g_wet_	150							
Inoculum, g_TS_	7.95							
Inoculum, g_VS_	0.00	4.71						
SIR by TS, -	-	0.38						
SIR by VS, -	-	0.61						
Additive material, g_TS_	-	-	0.03	0.09	0.145	0.20	0.26	0.35
Additive material, g_TS_ × L^−1^	-	-	0.20	0.60	0.95	1.34	1.70	2.30
Additive material to a substrate by TS, %	-	-	1	3	5	7	9	12

* dry substrate was placed into the reactors.

**Table 2 materials-16-05245-t002:** Comparison of additive materials effects on the anaerobic digestion of different organic wastes.

Substrate Type	Wet Mass of Substrate, g	Dry Mass of Substrate, g	Volatile Solids of a Substrate, g	Inoculum Type	Wet Mass of Inoculum, g	Dry Mass of Inoculum, g	Volatile Solids of Inoculum, g	Type of Additive Materials	Mass of Additive Materials, g × L^−1^	Reactors Type	Stirring	Total Volume, ml	Working Volume, mL	AD Temperature, °C	AD Duration Time, day	SIR by TS, -	SIR by VS, -	Main Effects/Findings	References
Brewer’s spent grain	11.7	3.0	2.88	Agriculture biogas plant digestate	150	7.95	4.71	Iron powder	0, 0.2, 0.6, 0.95, 1.34, 1.70, 2.30	Batch	Yes	400	153	37	30	0.38	0.61	× Fe supplementation increased methane yield in each experiment; the highest increase of 9.8% was obtained at doses of 0.6 0.6 g × L^−1^	This study
Dairy manure	600	54.0	44.3	Dairy manure	n.a	n.a	n.a	Iron powder	0, 2, 3, 4, 8, 12, 20	Batch	n.i	1000	600	38	48	n.a	n.a	× No effect on methane yield, × Reduction of H_2_S in biogas by 93–99%, × Increase in hydrolysis rate constant two-fold, × Lag-phase reduced to half	[36]
Waste activated sludge	n.i	n.i	n.i	Waste activated sludge	n.a	n.a	n.a	Iron powder	0, 1.0, 4.0	Batch	n.i	n.i	n.i	35	20	n.i	n.i	× Significant methane yield increase by 9 and 21% × No significant effect on k value	[37]
Waste activated sludge	n.i	n.i	n.i	Waste activated sludge	n.a	n.a	n.a	Iron powder, clean scrap, rusty scrap	0, 10	Batch	n.i	n.i	n.i	35	20	n.i	n.i	× Significant methane yield increase by 11, 22, and 30% × No significant effect on k value	[37]
Dewatered sludge	n.i	n.i	n.i	Anaerobically digested sludge	n.i	n.i	n.i	Iron powder	1.34	Batch	Yes	2000	1400	37	21	n.i	n.i	× Increase in methane yield by 18.15%, × Reduction of the diversity in the archaeal community, × Promotion of the growth of hydrogenotrophic methanogens, × Enhance in transportation and metabolism of carbohydrates and lipids by the archaeal community	[38]
Artificial wastewater	n.i	n.i	n.i	Seed sludge	n.i	TSS 12.9 g × L^−1^	VSS 3.1 g × L^−1^	Iron powder	0, 0.01565, 0.02108, 0.02899	Continuous	n.i	n.i	2000	35	95	n.a	n.a	× Enhance propionate conversion, × Increase of acetate production and COD removal, × Reduction of H_2_ and acceleration of homoacetogenesis × Accumulation of propionate could be alleviated by accelerating the conversion of propionate by iron powder	[22]
Brewer’s spent grain	11.7	3	2.88	Agriculture biogas plant digestate	150	7.95	4.71	Lime (Ca(OH)_2_)	0, 0.2, 0.6, 0.95, 1.34, 1.70, 2.31	Batch	Yes	400	153	37	30	0.38	0.61	× Lime supplementation decreased methane yield in each experiment by 3.3–6.7%	This study
Smooth cordgrass	200	80	n.i	Anaerobic seed cultures	210	11.3	5.6	Lime (Ca(OH)_2_)	0, 3, 6, 10	Batch, Leaching bed reactors	n.i	n.i	410	35	26	7.1	n.i	× Biogas yield inhibition by 7.1, 20, and 75.7%	[40]
Smooth cordgrass	1280	510	n.i	Anaerobic seed cultures	770	41.6	20.5	Lime (Ca(OH)_2_)	4, 7, 12	Leaching bed reactors	No	6000	2050	35	48	n.a	n.i	× Biogas yield decreased by 10.8, and 37% concerning the reactor with the lowest lime load, × Reduction of k value with increasing lime load 0.016, 0.011, and 0.005 d^−1^, × Methanogenic bacteria were more inhibited compared to other anaerobic bacteria	[40]
Food waste	50	10.1	2.0	Sewage sludge	100	4.0	2.3	Lime mud from the papermaking process (CaCO_3_, CaO)	0, 2, 6, 10, 14	Batch	n.i	800	500	37	40	2.53	0.89	× methane yield increase and enhancement of process stability, × Improved speed and balance of producing acid and methane, × Lime dosage increased organic substrate degradation, × doses of 2 and 14 g × L^−1^ show the lowest methane yield	[41]
Food waste	200	5.2	5.0	Sewage sludge	100	5.1	2.4	Lime mud from papermaking (LPM), waste eggshell (WES), CaCO_3_, NaHCO_3_	0, 8	Batch	n.i	1000	500	37	36	1.02	2.12	× improved pH-buffering capacity and inhibition alleviation of methanogenic process, × When there is a simultaneous presence of an alkalinity source and micronutrients, it enhances the stability of the process. × Buffer capacity from largest to smallest are LMP, CaCO_3_, WES, NaHCO_3_	[26]
Municipal solid waste (MSW)	7500	n.i	n.i	Leachate from MSW landfill	n.i	n.i	n.i	Na_2_CO_3_, NaHCO_3_, NaOH	n.i	Landfill simulated reactor	Leachate recalculation three times per week	n.i	n.i	25	80	n.a	n.a	Alkalinity addition: × had positive effects on the stabilization of MSW, × enhanced pH-buffering capacity and alleviates inhibition of methanogenesis, × Accelerated degradation rate of pollutants × had positive impacts on the transformation of nitrogen and total nitrogen removal	[43]
Brewer’s spent grain	11.7	3	2.88	Agriculture biogas plant digestate	150	7.95	4.71	Milled ceramic powder	0, 0.2, 0.6, 0.95, 1.34, 1.70, 2.32	Batch	Yes	400	153	37	30	0.38	0.61	× Ceramic powder did not significantly change methane yield	This study
Oily sludge	90	2.77	0.06	Oily sludge	n.a	n.a	n.a	Biochar	0, 5.6, 11,1, 22.2, 44.4	Batch	Yes	120	90	35	4	n.a	n.a	× 5.6 g × L^−1^ increased methane yield by 218%, × 44.4 g × L^−1^ decreased methane yield by 32.6%, × high surface area and abundant organic functional groups reshaped the microbial community, × Mitigation of biotoxicity suppression of oily sludge, × strengthening of microbial metabolism under BC added condition, × Strong adsorption of excessive BC (>5.6 g × L^−1^) inhibited mass transfer and caused negative effects on the AD process	[48]
× Piggery wastes, × Synthetic waste	n.i	n.i	n.i	Digested piggery waste	n.i	n.i	n.i	× Natural zeolite, × Sand	0, 0,05, 0.1, 0.15, 0.2, 0.25, 0.3, 0.4 g × g_VSS_^−1^	Batch	Yes	2500	n.i	27–31	40	n.i	n.i	× For piggery waste, the main mechanism of the AD enhancement was high microorganisms immobilization on zeolite, × For synthetic waste, the main mechanisms of the AD enhancement were high microorganisms immobilization and support for ammonia nitrogen reduction	[49]
Cattle manure processed under high ammonia and sulfate concentrations	n.i	n.i	n.i	Biogas plant digestate	n.i	n.i	n.i	× Bentonite, × Zeolite 13X, × Alkali-modified bentonites and zeolites	0, 8	Batch	Yes	321	150	37	35	n.i	1.0	× Zeolite 13X alleviates ammonia and sulfate co-derived toxicity, × Bentonite did not effectively mitigate the toxicity of ammonia and sulfate	[52]
Food waste	n.i	n.i	n.i	Inoculated sludge	n.i	n.i	n.i	× Fe-metal organic frameworks (Fe-MOF), × Ketjen Black (KB)	× 0, 0.25, 0.5, 0.75, 1, 1.25 (Fe-MOF), × 0, 0.1, 0.2, 0.3, 0.4, 0.5 (KB)	Batch	Yes	400	n.i	36	25	n.i	1.0	× addition of 0.5 g × L^−1^ of Fe-MOF increased methane yield by 27.5%, and shortened lag phase by 34.1%, × addition of 0.2 g × L^−1^ of KB increased methane yield by 29.5%, and shortened lag phase by 49.2%, × Fe-MOF and KB promote the activity of the electron transfer system up to two-fold, × functional groups (−OH, C=O, C=C, and −NH) can increase the buffering capacity of the digestive system	[53]
Brewer’s spent grain	n.i	4.73–5.25	4.56–5.07	Agriculture biogas plant digestate	30	0.93	0.56	Biochar made from BSG at 300 °C	0, 1.5, 4.5, 7.6, 12.1, 15.2	Batch	Yes	1000	35	37	21	5.7–5.12	7.03–7.81	× 4.5 g × L^−1^ increased biogas yield by 145% and biogas production constant by 30%, × overdosing with biochar (>12.1 g × L^−1^) decrease biogas yield	[2]
Brewer’s spent grain	10	n.i	n.i	Anaerobic sludge	390	n.i	n.i	× Biochar made from BSG at 300 °C, red spruce woodchips at 500 °C, × Granular activated carbon	around 4.5	Batch	Yes	650	400	35	19	n.i	0.167	× Depending on BSG type, the methane yield was improved by 26.6% and acidification was alleviated, or the AD process was inhibited after 7 days and methane yield decreased by 5%	[28]
Food waste	76	29.0	28.1	Granular sludge	74	n.i	n.i	Biochars made from BSG, food waste, and wood waste	0, 1.3, 2, 3.3, 5, 8	× Batch, × Continuous up-flow anaerobic sludge blanket reactor (UASB)	Yes	250	150	30	6	n.i	1	× biogas volume produced biochar was lower than the amount of biogas produced by the control with only food waste, × type of biochar and trace elements concentration in biochar plays a key role in determining the effectiveness of the biochar in enhancing biogas production from food waste, × Biochar enhanced the COD removal efficiency in the test UASB reactor by 37–47%.	[54]
Brewer’s spent grain	16.2	3.3	3.2	Agriculture biogas plant digestate	200	13.6	6.36	Biochars made from BSG at 300, 450, and 600 °C	0.3–31	Batch	Yes	400	203.3	37	30	0.24, 0.49, 0.97	0.5, 1.0, 2.0	× Biochar supplementation increased methane yield by 1.8, 10, and 3.1 for SIR 0.5, 1.0, and 2.0, respectively	[27]
Food waste	n.i	1.6–4.0	n.i	Sludge	n.i	3.2	n.i	Biochar	0, 2, 5, 10	Batch	n.i	1000	400	35	200	n.i	0.5, 1.0, 1.25	× Biochar reduced lag phase by 10–20% at SIR 0.5, by 43–54% at SIR 1.0, and by 36.3–54 at SIR 1.25, × Biochar increased methane yield by 100–275% at SIR 0.5, by 100–133 at SIR 1.0, and by 33–100 at SIR 1.25, × The effectiveness of biochar depends on the amount of biochar added and the amount of inoculum used.	[55]

n.i—no information, n.a—not applicable.

**Table 3 materials-16-05245-t003:** Kinetics parameters and substrate conversion efficiency.

Variant, -	*y_max_*, mL × g_VS_^−1^	*k*, d^−1^	*r*, mL × (g_VS_ × d)^−1^	*t*_1/2_, d	*BD*, %	*VSr*, %
Fe0	383.6 ± 2.6	0.07 ± 0.00	26.4 ± 1.5	10.1 ± 0.5	62.0 ± 2.1	43.0 ± 0.9
Fe1	389.1 ± 9.9	0.07 ± 0.00	28.3 ± 0.4	9.5 ± 0.4	63.9 ± 1.8	42.6 ± 0.6
Fe3	403.1 ± 9.2	0.08 ± 0.01	31.7 ± 1.5	8.8 ± 0.6	68.1 ± 0.7	43.6 ± 0.5
Fe5	360.3 ± 1.6	0.08 ± 0.00	29.1 ± 1.7	8.6 ± 0.5	60.6 ± 2.0	44.2 ± 1.9
Fe7	373.2 ± 0.1	0.08 ± 0.00	29.3 ± 0.9	8.8 ± 0.3	62.2 ± 0.7	43.4 ± 0.7
Fe9	383.4 ± 4.5	0.07 ± 0.00	26.8 ± 0.6	9.9 ± 0.1	61.5 ± 1.3	44.3 ± 1.5
Fe12	378.6 ± 5.3	0.07 ± 0.01	27.8 ± 1.6	9.5 ± 0.7	61.6 ± 0.7	40.7 ± 0.7
L0	320.2 ± 7.2	0.12 ± 0.01	40.1 ± 5.2	5.6 ± 0.6	58.8 ± 3.3	44.7 ± 0.0
L1	305.0 ± 2.0	0.11 ± 0.01	33.9 ± 3.0	6.3 ± 0.5	54.8 ± 1.6	45.6 ± 0.8
L3	306.6 ± 3.0	0.12 ± 0.00	37.2 ± 0.6	5.7 ± 0.0	55.9 ± 0.8	45.1 ± 1.3
L5	315.2 ± 5.3	0.11 ± 0.01	35.5 ± 1.0	6.2 ± 0.3	56.9 ± 0.7	42.9 ± 1.1
L7	314.4 ± 4.9	0.10 ± 0.00	32.3 ± 1.1	6.8 ± 0.1	55.6 ± 1.7	44.4 ± 0.0
L9	329.5 ± 13	0.11 ± 0.01	35.5 ± 0.8	6.4 ± 0.4	58.9 ± 2.3	44.8 ± 0.3
L12	316.7 ± 0.3	0.10 ± 0.01	32.5 ± 2.7	6.8 ± 0.6	56.0 ± 1.6	45.2 ± 0.1
Cf0	298.7 ± 1.8	0.24 ± 0.01	71.7 ± 2.5	2.9 ± 0.1	61.9 ± 0.7	44.7 ± 2.1
Cf1	296.2 ± 2.3	0.24 ± 0.01	70.6 ± 2.0	2.9 ± 0.1	61.3 ± 0.7	45.6 ± 1.0
Cf3	299.0 ± 6.7	0.24 ± 0.00	72.4 ± 2.9	2.9 ± 0.0	61.7 ± 1.8	45.1 ± 0.4
Cf5	296.0 ± 2.5	0.25 ± 0.01	73.3 ± 1.6	2.8 ± 0.1	61.1 ± 0.8	42.9 ± 0.8
Cf7	312.7 ± 6.7	0.22 ± 0.00	69.7 ± 2.9	3.1 ± 0.0	64.8 ± 6.2	44.4 ± 2.4
Cf9	292.7 ± 1.5	0.23 ± 0.01	68.1 ± 2.4	3.0 ± 0.1	60.3 ± 0.2	44.8 ± 0.7
Cf12	300.7 ± 2.0	0.23 ± 0.00	70.0 ± 1.0	3.0 ± 0.0	62.0 ± 0.7	45.2 ± 1.5

**Table 4 materials-16-05245-t004:** Process residue characteristics.

Variant, -	VS, %	C, %	H, %	N, %	S, %	O, %	AC, %	pH, -	EC, µS × cm^−1^
Fe0	63.7 ± 0.1	35.5 ± 1.1	5.0 ± 0.1	4.2 ± 0.3	1.4 ± 0.1	19.1 ± 0.4	36.3 ± 0.1	8.07 ± 0.01	33.80 ± 0.00
Fe1	64.2 ± 0.3	34.0 ± 0.4	4.5 ± 0.0	4.2 ± 0.2	1.3 ± 0.2	21.5 ± 0.2	35.8 ± 0.3	8.11 ± 0.01	34.50 ± 1.27
Fe3	62.8 ± 2.7	32.5 ± 0.9	4.2 ± 0.2	3.9 ± 0.2	1.4 ± 0.3	22.2 ± 0.4	37.2 ± 2.7	8.09 ± 0.01	34.70 ± 0.14
Fe5	62.8 ± 0.8	31.7 ± 2.1	4.0 ± 0.3	3.8 ± 0.1	1.5 ± 0.4	23.3 ± 0.7	37.2 ± 0.8	8.11 ± 0.02	34.30 ± 0.42
Fe7	63.9 ± 0.2	35.0 ± 1.3	4.5 ± 0.4	4.6 ± 0.5	1.5 ± 0.1	19.7 ± 0.6	36.1 ± 0.2	8.11 ± 0.01	34.20 ± 0.57
Fe9	63.0 ± 0.9	34.3 ± 0.4	4.3 ± 0.3	4.5 ± 0.1	1.4 ± 0.0	19.9 ± 0.2	37.0 ± 0.9	8.14 ± 0.00	34.45 ± 0.21
Fe12	64.5 ± 0.3	32.5 ± 1.6	4.2 ± 0.3	4.4 ± 0.6	1.4 ± 0.1	23.4 ± 0.6	35.5 ± 0.3	8.13 ± 0.00	34.70 ± 0.14
L0	62.4 ± 1.2	33.1 ± 2.3	4.3 ± 0.6	3.7 ± 0.1	1.2 ± 0.0	21.3 ± 0.7	37.6 ± 1.2	7.92 ± 0.02	34.95 ± 0.07
L1	63.4 ± 0.1	32.1 ± 0.3	4.1 ± 0.0	4.1 ± 0.2	1.2 ± 0.1	23.1 ± 0.2	36.6 ± 0.1	7.94 ± 0.00	35.70 ± 0.00
L3	62.5 ± 0.6	33.4 ± 1.5	4.3 ± 0.3	4.1 ± 0.6	1.3 ± 0.1	20.6 ± 0.6	37.5 ± 0.6	7.94 ± 0.00	33.50 ± 0.07
L5	62.3 ± 1.7	35.3 ± 1.1	4.4 ± 0.2	4.6 ± 0.6	1.2 ± 0.1	18.0 ± 0.5	37.7 ± 1.7	7.95 ± 0.04	34.80 ± 0.07
L7	63.1 ± 0.1	33.8 ± 1.0	4.4 ± 0.3	4.8 ± 0.0	1.2 ± 0.2	20.1 ± 0.4	36.9 ± 0.1	7.95 ± 0.01	35.15 ± 0.07
L9	61.6 ± 0.3	30.6 ± 1.8	3.9 ± 0.4	4.6 ± 0.4	1.1 ± 0.4	22.5 ± 0.7	38.4 ± 0.3	7.95 ± 0.01	35.30 ± 0.07
L12	61.0 ± 0.1	31.6 ± 0.7	3.9 ± 0.1	4.4 ± 0.0	0.8 ± 0.1	21.1 ± 0.2	39.0 ± 0.1	7.94 ± 0.01	35.50 ± 0.07
Cf0	63.4 ± 1.2	32.6 ± 0.3	4.3 ± 0.1	3.9 ± 0.1	1.1 ± 0.2	22.5 ± 0.2	36.6 ± 1.2	7.92 ± 0.02	34.95 ± 1.06
Cf1	63.8 ± 1.1	32.3 ± 0.5	4.2 ± 0.2	4.1 ± 0.0	1.2 ± 0.1	23.1 ± 0.2	36.2 ± 1.1	7.94 ± 0.02	35.70 ± 0.14
Cf3	63.6 ± 1.6	32.5 ± 2.6	4.3 ± 0.6	4.0 ± 0.4	1.1 ± 0.0	22.8 ± 0.9	36.4 ± 1.6	7.94 ± 0.03	33.50 ± 0.78
Cf5	63.6 ± 2.2	31.7 ± 5.1	4.1 ± 1.0	4.2 ± 0.3	1.0 ± 0.1	23.6 ± 1.6	36.4 ± 2.2	7.95 ± 0.01	34.80 ± 0.99
Cf7	63.7 ± 1.5	30.5 ± 0.7	4.1 ± 0.2	4.7 ± 0.3	1.0 ± 0.0	24.4 ± 0.3	36.3 ± 1.5	7.95 ± 0.03	35.15 ± 0.49
Cf9	62.8 ± 0.9	31.7 ± 0.1	4.2 ± 0.0	4.6 ± 0.2	1.1 ± 0.1	22.3 ± 0.1	37.2 ± 0.9	7.95 ± 0.04	35.30 ± 0.57
Cf12	62.1 ± 1.3	32.0 ± 0.2	4.0 ± 0.1	4.4 ± 0.3	1.0 ± 0.1	21.8 ± 0.2	37.9 ± 1.3	7.94 ± 0.01	35.50 ± 2.33

**Table 5 materials-16-05245-t005:** Micro- and macro-nutrients in process residues, mg × kg_TS_^−1^.

Share, %	Fe	Co	Mo	Se	W	Cu	Zn	Mn
Fe0	3300 ± 500	1.5 ± 0.3	1.6 ± 0.3	<5.0	<5.0	27.5 ± 5.5	210 ± 40	99 ± 20
Fe1	3350 ± 500	1.6 ± 0.3	1.2 ± 0.2	<5.0	<5.0	25.5 ± 5.5	195 ± 40	87 ± 18
Fe3	5400 ± 800	2.2 ± 0.5	1.6 ± 0.3	<5.0	<5.0	28.5 ± 5.5	220 ± 45	96 ± 19
Fe5	5650 ± 850	2.2 ± 0.5	1.8 ± 0.4	<5.0	<5.0	28.5 ± 6.0	215 ± 40	96.5 ± 19
Fe7	6950 ± 1050	2.8 ± 0.6	2.9 ± 0.6	<5.0	<5.0	31.5 ± 3.0	230 ± 45	102 ± 19
Fe9	6400 ± 950	2.6 ± 0.5	1.4 ± 0.3	<5.0	<5.0	26.5 ± 5.0	200 ± 40	86 ± 17
Fe12	8650 ± 1300	3.3 ± 0.7	2.0 ± 0.4	<5.0	<5.0	31.5 ± 6.5	240 ± 50	104 ± 35
L0	3500 ± 550	1.5 ± 0.3	1.7 ± 0.3	<5.0	<5.0	31.0 ± 6.0	190 ± 40	100 ± 30
L1	3550 ± 550	1.7 ± 0.3	2.2 ± 0.5	<5.0	<5.0	30.0 ± 6.0	200 ± 40	97 ± 19
L3	3350 ± 500	1.4 ± 0.3	2.3 ± 0.5	<5.0	<5.0	28.5 ± 5.5	180 ± 35	96 ± 29
L5	3150 ± 450	1.3 ± 0.3	1.9 ± 0.4	<5.0	<5.0	27.0 ± 5.5	175 ± 35	92 ± 18
L7	3800 ± 550	2.1 ± 0.4	2.1 ± 1.8	<5.0	<5.0	33.5 ± 6.5	195 ± 40	104 ± 20
L9	3550 ± 550	1.6 ± 0.3	2.3 ± 0.5	<5.0	<5.0	30.5 ± 6.0	205 ± 40	103 ± 19
L12	3500 ± 500	1.6 ± 0.3	2.2 ± 0.5	<5.0	<5.0	29.5 ± 6.0	195 ± 40	97 ± 19
Cf0	3520 ± 520	1.5 ± 0.5	1.6 ± 0.6	<5.0	<5.0	29.3 ± 7.5	200 ± 45	99 ± 25
Cf1	3450 ± 520	1.6 ± 0.6	1.7 ± 0.7	<5.0	<5.0	27.0 ± 8.5	198 ± 55	93 ± 35
Cf3	3250 ± 650	1.8 ± 0.6	1.9 ± 0.6	<5.0	<5.0	28.5 ± 8.5	200 ± 50	97 ± 30
Cf5	3400 ± 650	1.7 ± 0.6	1.8 ± 0.5	<5.0	<5.0	27.75 ± 8	195 ± 50	94 ± 30
Cf7	3680 ± 800	2.4 ± 0.5	2.5 ± 1.2	<5.0	<5.0	32.5 ± 7.5	213 ± 45	103 ± 30
Cf9	3530 ± 750	2.1 ± 0.5	1.8 ± 0.4	<5.0	<5.0	28.5 ± 8.0	203 ± 50	94 ± 30
Cf12	3520 ± 900	2.5 ± 0.6	2.1 ± 0.6	<5.0	<5.0	30.5 ± 8.0	218 ± 50	100 ± 30

## Data Availability

Essential data are presented in the manuscript in the Results Section.

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
