# Peer review of "Effects of Iron, Lime, and Porous Ceramic Powder Additives on Methane Production from Brewer’s Spent Grain in the Anaerobic Digestion Process"

_materials, 2023, doi:10.3390/ma16155245_

Round 1

Reviewer 1 Report

   The effects of iron powder, lime, and milled porous ceramic on anaerobic process and methane production were studied experimentally in the manuscript. The scientific problems of the article are clear, the experimental design is reasonable, the research results are fully discussed, and the writing is standardized.

1.     What is the main question addressed by the research?

The authors tried to increase the gas production rate by adding stimulating materials to the anaerobic process.

2. Do you consider the topic original or relevant in the field? Does it

address a specific gap in the field?

There are numerous research papers pertaining to anaerobic fermentation, many of which share similarities. The author posits that the innovation of this paper lies primarily in its choice of fermentation raw materials, while I believe that its innovation is more general in nature.

3. What does it add to the subject area compared with other published

material?

The addition of new content is limited, with the primary fermentation raw materials consisting of food waste and non-agricultural waste commonly found in the industry.

4. What specific improvements should the authors consider regarding the

methodology? What further controls should be considered?

It is recommended that the authors employ detection methods, such as fluorescence quantitative PCR (RT-PCR) technology, to investigate the impact of additives on microbial community changes and further elucidate their effects on gas production.

5. Are the conclusions consistent with the evidence and arguments presented

and do they address the main question posed?

The conclusion of the study is recommended to be revised, not merely reiterating the research findings. Instead, it should provide a clear verdict on how additives affect gas yield and describe their applicable conditions and significance for popularization.

6. Are the references appropriate?

The inclusion of research papers on anaerobic fermentation of industrial food waste is recommended.

7. Please include any additional comments on the tables and figures.

Several curves in FIG. 1-3 exhibit a high degree of overlap, which falls within the error bar range and does not directly reflect the impact of additives on gas yield. It is recommended to employ a table for quantitative expression. Furthermore, we suggest that the authors present a comparative analysis graph showcasing the results of anaerobic fermentation with and without three different additives.

  It is suggested to supplement the theoretical value, practical value and application scope of the research results in the conclusion section. It is suggested that the article can be published after minor revisions.

Reviewer 2 Report

The manuscript has merit. However, some corrections are needed:

Table 1. Once "Inoculum, gwet and Inoculum, gTS" is the same value in all experiments, this information should be included in the description and removed from the table.

Discussion: Some section (table and text) comparing the best results of this work with those present in the literature must be added.

Reviewer 3 Report

The article “Effects of Stimulating Materials Addition on Methane Production from Brewer’s Spent Grain in Anaerobic Digestion Process” deals with the addition of materials to the anaerobic digestion process of spent grain. The article is in the scope of Materials and is structured in a good way. However, its novelty appears limited. Furthermore, there are several aspects to refine which require a major revision; in detail, the authors should consider these aspects:

1.      Introduction: Recently, biochar addition to BSG digestion has been tested as well. Biochar is an interesting carbonaceous additive that can be used instead of conventional activated carbon; this possibility should also be mentioned in the section. See 10.1016/j.jece.2019.103184 and 10.3390/en12081518 as relevant references.

2.      Introduction: the novelty of the manuscript must be better highlighted when compared to the existing literature on the topic.

3.      Section 2.1: why was BSG dried before AD? Normally wet BSG is used, as this can also undermine the energy balance requiring significant energy inputs.

4.      Section 2.1: It is unclear why those specific additives were selected, among all possible additives tested in the literature.

5.      Lines 172-174: the verb is repeated twice.

6.      Line 184-185: how were the different ratios of additive to BSG chosen?

7.      Please check Equation 3, it looks wrong.

8.      Lines 212-216: please check the equations numbers mentioned in the text.

9.      Section 3.1: A Table should be reported summarizing the physicochemical characteristics of the materials, instead of reporting only written text (lines 233-241).

10.   Fig. 1-4: sometimes it is difficult to appreciate the difference between the curves. I suggest using also dotted lines beside colors to better discriminate the curves.

11.   Line 316: please avoid abbreviated forms such as “what’s” or “weren’t”.

12.   Section 3.2: The presentation of the results is good, but the discussion with the literature is too extensive. Please reduce the length of the section at least by 20%.

13.   Section 3.2: I miss a (basic) statistical analysis of the results, which is mentioned only for the results reported in Fig. 4. Are the results (CH4 increase or decrease) statistically significant or not?

14.   Section 3.3: besides analyzing the digestate quality, a comparison with the legislation standards for agricultural reuse should be done. E.g., it could be interesting to understand if after Fe addition the digestate is still suitable to be applied to the fields (or not). Consider 10.3390/su11216015 and 10.31025/2611-4135/2020.13993 as pertinent references on these aspects.

15.   Section 3.3: some more details about the future studies should be reported, apart from what is stated in the Conclusions.

16.   The English language is good and does not need significant revision, besides some punctual errors. 

The English language is good and does not need significant revision, besides some punctual errors. 

Reviewer 4 Report

Manuscript: materials-2476705-peer-review-v1

Title: “Effects of Stimulating Materials Addition on Methane Production from Brewer’s Spent Grain in Anaerobic Digestion Process”

Amid the very lengthy writing, the findings of the paper can be summed up in a short statement: the proposed digestion enhancement technique did not really work. Hence, the first sentence in the Abstract does not represent the findings: “The process of anaerobic digestion used for methane production can be enhanced by incorporating stimulating materials.”

And the meaning of the terms “stimulation”, “stimulating”, “stimulate” is not really a new concept or technique. Anything that promotes microbial growth is a stimulator of bio-growth. So, in essence, “stimulation” etc. are describing a very traditional and fundamental concept of any microbial growth, which is the provision of the substrates and nutrients needed by the microbes to grow. This makes the paper not new; rather, the paper contains a traditional exercise managing microbial systems. If there is really a unique mechanism in the concept of “stimulation”, that is not clear in the paper. And this must be cleared because it is the foundation of merit on innovation or contribution to the literature.

Also, there are so many tables that contain data better suited to graphical rendering for easier analysis of trends. The reader sees trends better in graphs than by sifting through values in tables.

Minor editing needed.

Round 2

Reviewer 3 Report

The authors addressed in a good way reviewer comments. The manuscript can thus be published.

The English language is good.

Author Response

Thank you.

Reviewer 4 Report

Issue on the technique they claim as “stimulation”: as pointed out in the 1st review, it is crucial that the paper clearly distinguishes “stimulation” from the typical “cultivation” of microbial systems. They keep on using the concept “stimulation” using various version of the word in the title and throughout the paper like it is a clearly established concept, but there is no clear established concept of “stimulation” in their experimental setup.

Issue on the fit of the subject of the paper to the Materials MDPI journal: Even though the paper discusses works on the use of various materials including porous ceramics, iron powder, and lime, the whole is about techniques in the area of fermentation and bioprocessing, which are not the focus of Materials MDPI.

Issue on experimental design: the paper claims that it is bringing a novelty into the literature by examining “mix” loading of various substances as “stimulants” of anaerobic digestion, but the experimental design is not a correct “mixture design”. I cannot even identify what kind of statistical experimental design they followed to generate the plan of experiments shown in Table 1.

Issue on kinetics model development: the computations from raw data to the model parameters were not clear. Simply stating “using least square methods employing Statistica software” is not enough. For example, how was equation 1 derived from first principle of production mass rate, from the differential form to the integral form? My own derivations based on classical rate of reactions do no result to the same equations they show in the paper. Show the details even by moving such details in the Appendix section. Also, how was the rate of methane production “r” calculated from the raw data?

The paper, after the modifications from review round 1, now sounds like less clear on its purpose because new materials were added amid not having a clear thesis of the paper. Is this work really about the claimed “stimulation”? or just a typical microbial cultivation and it just happens that the substances used in this work are also popular materials in other research area? Is the work’s novelty as claimed in the introduction really on “mixture” of the “stimulant” substances? If so, then follow a correct mixture design and make sure to mention the name of that mixture design of experiment so it is clear in the paper.

Extensive editing of English language required.
